# Frank-Wolfe Bayesian Quadrature: Probabilistic Integration with Theoretical Guarantees

**François-Xavier Briol**
Department of Statistics
University of Warwick
f-x.briol@warwick.ac.uk

**Chris J. Oates**
School of Mathematical and Physical Sciences
University of Technology, Sydney
christopher.oates@uts.edu.au

**Mark Girolami**
Department of Statistics
University of Warwick
m.girolami@warwick.ac.uk

**Michael A. Osborne**
Department of Engineering Science
University of Oxford
mosb@robots.ox.ac.uk

## Abstract

There is renewed interest in formulating integration as a statistical inference problem, motivated by obtaining a full distribution over numerical error that can be propagated through subsequent computation. Current methods, such as Bayesian Quadrature, demonstrate impressive empirical performance but lack theoretical analysis. An important challenge is therefore to reconcile these probabilistic integrators with rigorous convergence guarantees. In this paper, we present the first probabilistic integrator that admits such theoretical treatment, called Frank-Wolfe Bayesian Quadrature (FWBQ). Under FWBQ, convergence to the true value of the integral is shown to be up to exponential and posterior contraction rates are proven to be up to super-exponential. In simulations, FWBQ is competitive with state-of-the-art methods and out-performs alternatives based on Frank-Wolfe optimisation. Our approach is applied to successfully quantify numerical error in the solution to a challenging Bayesian model choice problem in cellular biology.

## 1 Introduction

Computing integrals is a core challenge in machine learning and numerical methods play a central role in this area. This can be problematic when a numerical integration routine is repeatedly called, maybe millions of times, within a larger computational pipeline. In such situations, the cumulative impact of numerical errors can be unclear, especially in cases where the error has a non-trivial structural component. One solution is to model the numerical error statistically and to propagate this source of uncertainty through subsequent computations. Conversely, an understanding of how errors arise and propagate can enable the efficient focusing of computational resources upon the most challenging numerical integrals in a pipeline.

Classical numerical integration schemes do not account for prior information on the integrand and, as a consequence, can require an excessive number of function evaluations to obtain a prescribed level of accuracy [21]. Alternatives such as Quasi-Monte Carlo (QMC) can exploit knowledge on the smoothness of the integrand to obtain optimal convergence rates [7]. However these optimal rates can only hold on sub-sequences of sample sizes $n$, a consequence of the fact that all function evaluations are weighted equally in the estimator [24]. A modern approach that avoids this problem is to consider arbitrarily weighted combinations of function values; the so-called *quadrature rules* (also called cubature rules). Whilst quadrature rules with non-equal weights have received comparatively little theoretical attention, it is known that the extra flexibility given by arbitrary weights can

lead to extremely accurate approximations in many settings (see applications to image de-noising [3] and mental simulation in psychology [13]).

Probabilistic numerics, introduced in the seminal paper of [6], aims at re-interpreting numerical tasks as inference tasks that are amenable to statistical analysis.[1] Recent developments include probabilistic solvers for linear systems [14] and differential equations [5, 26]. For the task of computing integrals, Bayesian Quadrature (BQ) [22] and more recent work by [20] provide probabilistic numerics methods that produce a full posterior distribution on the output of numerical schemes. One advantage of this approach is that we can propagate uncertainty through all subsequent computations to explicitly model the impact of numerical error [15]. Contrast this with chaining together classical error bounds; the result in such cases will typically be a weak bound that provides no insight into the error structure. At present, a significant shortcoming of these methods is the absence of theoretical results relating to rates of posterior contraction. This is unsatisfying and has likely hindered the adoption of probabilistic approaches to integration, since it is not clear that the induced posteriors represent a sensible quantification of the numerical error (by classical, frequentist standards).

This paper establishes convergence rates for a new probabilistic approach to integration. Our results thus overcome a key perceived weakness associated with probabilistic numerics in the quadrature setting. Our starting point is recent work by [2], who cast the design of quadrature rules as a problem in convex optimisation that can be solved using the Frank-Wolfe (FW) algorithm. We propose a hybrid approach of [2] with BQ, taking the form of a quadrature rule, that (i) carries a full probabilistic interpretation, (ii) is amenable to rigorous theoretical analysis, and (iii) converges orders-of-magnitude faster, empirically, compared with the original approaches in [2]. In particular, we prove that super-exponential rates hold for posterior contraction (concentration of the posterior probability mass on the true value of the integral), showing that the posterior distribution provides a sensible and effective quantification of the uncertainty arising from numerical error. The methodology is explored in simulations and also applied to a challenging model selection problem from cellular biology, where numerical error could lead to mis-allocation of expensive resources.

## 2 Background

### 2.1 Quadrature and Cubature Methods

Let $\mathcal{X} \subseteq \mathbb{R}^d$ be a measurable space such that $d \in \mathbb{N}_+$ and consider a probability density $p(x)$ defined with respect to the Lebesgue measure on $\mathcal{X}$. This paper focuses on computing integrals of the form $\int f(x)p(x)\mathrm{d}x$ for a test function $f : \mathcal{X} \to \mathbb{R}$ where, for simplicity, we assume $f$ is square-integrable with respect to $p(x)$. A *quadrature rule* approximates such integrals as a weighted sum of function values at some design points $\{x_i\}_{i=1}^n \subset \mathcal{X}$:

$$\int_{\mathcal{X}} f(x)p(x)\mathrm{d}x \approx \sum_{i=1}^n w_i f(x_i). \tag{1}$$

Viewing integrals as projections, we write $p[f]$ for the left-hand side and $\hat{p}[f]$ for the right-hand side, where $\hat{p} = \sum_{i=1}^n w_i \delta(x_i)$ and $\delta(x_i)$ is a Dirac measure at $x_i$. Note that $\hat{p}$ may not be a probability distribution; in fact, weights $\{w_i\}_{i=1}^n$ do not have to sum to one or be non-negative. Quadrature rules can be extended to multivariate functions $f : \mathcal{X} \to \mathbb{R}^d$ by taking each component in turn.

There are many ways of choosing combinations $\{x_i, w_i\}_{i=1}^n$ in the literature. For example, taking weights to be $w_i = 1/n$ with points $\{x_i\}_{i=1}^n$ drawn independently from the probability distribution $p(x)$ recovers basic Monte Carlo integration. The case with weights $w_i = 1/n$, but with points chosen with respect to some specific (possibly deterministic) schemes includes kernel herding [4] and Quasi-Monte Carlo (QMC) [7]. In Bayesian Quadrature, the points $\{x_i\}_{i=1}^n$ are chosen to minimise a posterior variance, with weights $\{w_i\}_{i=1}^n$ arising from a posterior probability distribution.

Classical error analysis for quadrature rules is naturally couched in terms of minimising the worst-case estimation error. Let $\mathcal{H}$ be a Hilbert space of functions $f : \mathcal{X} \to \mathbb{R}$, equipped with the inner

product $\langle \cdot, \cdot \rangle_{\mathcal{H}}$ and associated norm $\| \cdot \|_{\mathcal{H}}$. We define the *maximum mean discrepancy* (MMD) as:

$$\text{MMD}\big(\{x_i, w_i\}_{i=1}^n\big) := \sup_{f \in \mathcal{H}: \|f\|_{\mathcal{H}} = 1} \big| p[f] - \hat{p}[f] \big|. \tag{2}$$

The reader can refer to [27] for conditions on $\mathcal{H}$ that are needed for the existence of the MMD. The rate at which the MMD decreases with the number of samples $n$ is referred to as the 'convergence rate' of the quadrature rule. For Monte Carlo, the MMD decreases with the slow rate of $\mathcal{O}_P(n^{-1/2})$ (where the subscript $P$ specifies that the convergence is in probability). Let $\mathcal{H}$ be a RKHS with reproducing kernel $k : \mathcal{X} \times \mathcal{X} \to \mathbb{R}$ and denote the corresponding canonical feature map by $\Phi(x) = k(\cdot, x)$, so that the mean element is given by $\mu_p(x) = p[\Phi(x)] \in \mathcal{H}$. Then, following [27]

$$\text{MMD}\big(\{x_i, w_i\}_{i=1}^n\big) = \|\mu_p - \mu_{\hat{p}}\|_{\mathcal{H}}. \tag{3}$$

This shows that to obtain low integration error in the RKHS $\mathcal{H}$, one only needs to obtain a good approximation of its mean element $\mu_p$ (as $\forall f \in \mathcal{H}$: $p[f] = \langle f, \mu_p \rangle_{\mathcal{H}}$). Establishing theoretical results for such quadrature rules is an active area of research [1].

## 2.2 Bayesian Quadrature

Bayesian Quadrature (BQ) was originally introduced in [22] and later revisited by [11, 12] and [23]. The main idea is to place a functional prior on the integrand $f$, then update this prior through Bayes' theorem by conditioning on both samples $\{x_i\}_{i=1}^n$ and function evaluations at those sample points $\{f_i\}_{i=1}^n$ where $f_i = f(x_i)$. This induces a full posterior distribution over functions $f$ and hence over the value of the integral $p[f]$. The most common implementation assumes a Gaussian Process (GP) prior $f \sim \mathcal{GP}(0, k)$. A useful property motivating the use of GPs is that linear projection preserves normality, so that the posterior distribution for the integral $p[f]$ is also a Gaussian, characterised by its mean and covariance. A natural estimate of the integral $p[f]$ is given by the mean of this posterior distribution, which can be compactly written as

$$\hat{p}_{\text{BQ}}[f] = z^T K^{-1} f. \tag{4}$$

where $z_i = \mu_p(x_i)$ and $K_{ij} = k(x_i, x_j)$. Notice that this estimator takes the form of a quadrature rule with weights $w^{\text{BQ}} = z^T K^{-1}$. Recently, [25] showed how specific choices of kernel and design points for BQ can recover classical quadrature rules. This begs the question of how to select design points $\{x_i\}_{i=1}^n$. A particularly natural approach aims to minimise the posterior uncertainty over the integral $p[f]$, which was shown in [16, Prop. 1] to equal:

$$v_{\text{BQ}}\big(\{x_i\}_{i=1}^n\big) \;=\; p[\mu_p] - z^T K^{-1} z \;=\; \text{MMD}^2\big(\{x_i, w_i^{\text{BQ}}\}_{i=1}^n\big). \tag{5}$$

Thus, in the RKHS setting, minimising the posterior variance corresponds to minimising the worst case error of the quadrature rule. Below we refer to Optimal BQ (OBQ) as BQ coupled with design points $\{x_i^{\text{OBQ}}\}_{i=1}^n$ chosen to globally minimise (5). We also call Sequential BQ (SBQ) the algorithm that greedily selects design points to give the greatest decrease in posterior variance at each iteration. OBQ will give improved results over SBQ, but cannot be implemented in general, whereas SBQ is comparatively straight-forward to implement. There are currently no theoretical results establishing the convergence of either BQ, OBQ or SBQ.

*Remark:* (5) is independent of observed function values $f$. As such, no active learning is possible in SBQ (i.e. surprising function values never cause a revision of a planned sampling schedule). This is not always the case: For example [12] approximately encodes non-negativity of $f$ into BQ which leads to a dependence on $f$ in the posterior variance. In this case sequential selection becomes an *active* strategy that outperforms batch selection in general.

## 2.3 Deriving Quadrature Rules via the Frank-Wolfe Algorithm

Despite the elegance of BQ, its convergence rates have not yet been rigorously established. In brief, this is because $\hat{p}_{\text{BQ}}[f]$ is an orthogonal projection of $f$ onto the *affine* hull of $\{\Phi(x_i)\}_{i=1}^n$, rather than e.g. the *convex* hull. Standard results from the optimisation literature apply to bounded domains, but the affine hull is not bounded (i.e. the BQ weights can be arbitrarily large and possibly negative). Below we describe a solution to the problem of computing integrals recently proposed by [2], based on the FW algorithm, that restricts attention to the (bounded) convex hull of $\{\Phi(x_i)\}_{i=1}^n$.

**Algorithm 1** The Frank-Wolfe (FW) and Frank-Wolfe with Line-Search (FWLS) Algorithms.

---

**Require:** function $J$, initial state $g_1 = \bar{g}_1 \in \mathcal{G}$ (and, for FW only: step-size sequence $\{\rho_i\}_{i=1}^n$).

1: **for** $i = 2, \ldots, n$ **do**
2:     Compute $\bar{g}_i = \operatorname{argmin}_{g \in \mathcal{G}} \langle g, (DJ)(g_{i-1}) \rangle_\times$
3:     [For FWLS only, line search: $\rho_i = \operatorname{argmin}_{\rho \in [0,1]} J\big((1-\rho)g_{i-1} + \rho\,\bar{g}_i\big)$]
4:     Update $g_i = (1 - \rho_i)g_{i-1} + \rho_i \bar{g}_i$
5: **end for**

---

The Frank-Wolfe (FW) algorithm (Alg. 1), also called the conditional gradient algorithm, is a convex optimization method introduced in [9]. It considers problems of the form $\min_{g \in \mathcal{G}} J(g)$ where the function $J : \mathcal{G} \to \mathbb{R}$ is convex and continuously differentiable. A particular case of interest in this paper will be when the domain $\mathcal{G}$ is a compact and convex space of functions, as recently investigated in [17]. These assumptions imply the existence of a solution to the optimization problem.

In order to define the algorithm rigorously in this case, we introduce the Fréchet derivative of $J$, denoted $DJ$, such that for $\mathcal{H}^*$ being the dual space of $\mathcal{H}$, we have the unique map $DJ : \mathcal{H} \to \mathcal{H}^*$ such that for each $g \in \mathcal{H}$, $(DJ)(g)$ is the function mapping $h \in \mathcal{H}$ to $(DJ)(g)(h) = \langle g - \mu, h \rangle_{\mathcal{H}}$. We also introduce the bilinear map $\langle \cdot, \cdot \rangle_\times : \mathcal{H} \times \mathcal{H}^* \to \mathbb{R}$ which, for $F \in \mathcal{H}^*$ given by $F(g) = \langle g, f \rangle_{\mathcal{H}}$, is the rule giving $\langle h, F \rangle_\times = \langle h, f \rangle_{\mathcal{H}}$.

At each iteration $i$, the FW algorithm computes a linearisation of the objective function $J$ at the previous state $g_{i-1} \in \mathcal{G}$ along its gradient $(DJ)(g_{i-1})$ and selects an 'atom' $\bar{g}_i \in \mathcal{G}$ that minimises the inner product a state $g$ and $(DJ)(g_{i-1})$. The new state $g_i \in \mathcal{G}$ is then a convex combination of the previous state $g_{i-1}$ and of the atom $\bar{g}_i$. This convex combination depends on a step-size $\rho_i$ which is pre-determined and different versions of the algorithm may have different step-size sequences.

Our goal in quadrature is to approximate the mean element $\mu_p$. Recently [2] proposed to frame integration as a FW optimisation problem. Here, the domain $\mathcal{G} \subseteq \mathcal{H}$ is a space of functions and taking the objective function to be:

$$J(g) = \frac{1}{2} \big\| g - \mu_p \big\|_{\mathcal{H}}^2. \tag{6}$$

This gives an approximation of the mean element and $J$ takes the form of half the posterior variance (or the MMD$^2$). In this functional approximation setting, minimisation of $J$ is carried out over $\mathcal{G} = \mathcal{M}$, the marginal polytope of the RKHS $\mathcal{H}$. The marginal polytope $\mathcal{M}$ is defined as the closure of the convex hull of $\Phi(\mathcal{X})$, so that in particular $\mu_p \in \mathcal{M}$. Assuming as in [18] that $\Phi(x)$ is uniformly bounded in feature space (i.e. $\exists R > 0 : \forall x \in \mathcal{X}, \|\Phi(x)\|_{\mathcal{H}} \leq R$), then $\mathcal{M}$ is a closed and bounded set and can be optimised over.

A particular advantage of this method is that it leads to 'sparse' solutions which are linear combinations of the atoms $\{\bar{g}_i\}_{i=1}^n$ [2]. In particular this provides a weighted estimate for the mean element:

$$\hat{\mu}_{\text{FW}} := g_n = \sum_{i=1}^n \Big( \prod_{j=i+1}^n \big(1 - \rho_{j-1}\big)\rho_{i-1} \Big) \bar{g}_i := \sum_{i=1}^n w_i^{\text{FW}} \bar{g}_i, \tag{7}$$

where by default $\rho_0 = 1$ which leads to all $w_i^{\text{FW}} \in [0,1]$ when $\rho_i = 1/(i+1)$. A typical sequence of approximations to the mean element is shown in Fig. 1 (left), demonstrating that the approximation quickly converges to the ground truth (in black). Since minimisation of a linear function can be restricted to extreme points of the domain, the atoms will be of the form $\bar{g}_i = \Phi(x_i^{\text{FW}}) = k(\cdot, x_i^{\text{FW}})$ for some $x_i^{\text{FW}} \in \mathcal{X}$. The minimisation in $g$ over $\mathcal{G}$ from step 2 in Algorithm 1 therefore becomes a minimisation in $x$ over $\mathcal{X}$ and this algorithm therefore provides us design points. In practice, at each iteration $i$, the FW algorithm hence selects a design point $x_i^{\text{FW}} \in \mathcal{X}$ which induces an atom $\bar{g}_i$ and gives us an approximation of the mean element $\mu_p$. We denote by $\hat{\mu}_{\text{FW}}$ this approximation after $n$ iterations. Using the reproducing property, we can show that the FW estimate is a quadrature rule:

$$\hat{p}_{\text{FW}}[f] := \big\langle f, \hat{\mu}_{\text{FW}} \big\rangle_{\mathcal{H}} = \Big\langle f, \sum_{i=1}^n w_i^{\text{FW}} \bar{g}_i \Big\rangle_{\mathcal{H}} = \sum_{i=1}^n w_i^{\text{FW}} \big\langle f, k(\cdot, x_i^{\text{FW}}) \big\rangle_{\mathcal{H}} = \sum_{i=1}^n w_i^{\text{FW}} f(x_i^{\text{FW}}). \tag{8}$$

The total computational cost for FW is $\mathcal{O}(n^2)$. An extension known as FW with Line Search (FWLS) uses a line-search method to find the optimal step size $\rho_i$ at each iteration (see Alg. 1).

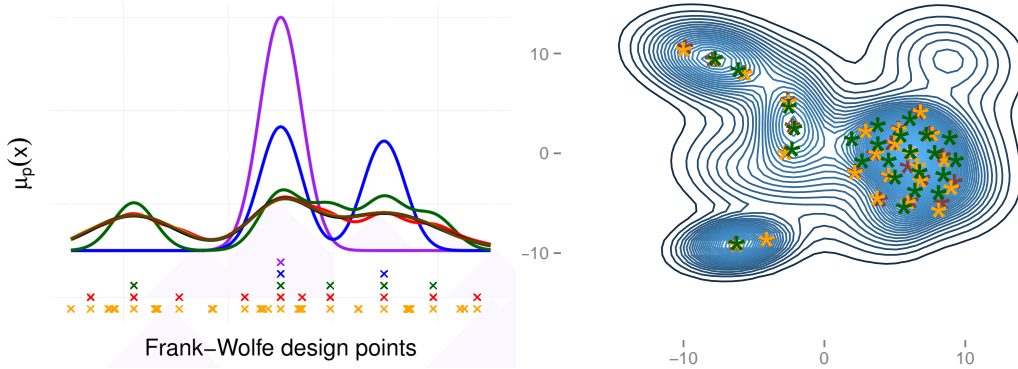

Figure 1: *Left*: Approximations of the mean element $\mu_p$ using the FWLS algorithm, based on $n = 1, 2, 5, 10, 50$ design points (purple, blue, green, red and orange respectively). It is not possible to distinguish between approximation and ground truth when $n = 50$. *Right*: Density of a mixture of 20 Gaussian distributions, displaying the first $n = 25$ design points chosen by FW (red), FWLS (orange) and SBQ (green). Each method provides well-spaced design points in high-density regions. Most FW and FWLS design points overlap, partly explaining their similar performance in this case.

Once again, the approximation obtained by FWLS has a sparse expression as a convex combination of all the previously visited states and we obtain an associated quadrature rule. FWLS has theoretical convergence rates that can be stronger than standard versions of FW but has computational cost in $\mathcal{O}(n^3)$. The authors in [10] provide a survey of FW-based algorithms and their convergence rates under different regularity conditions on the objective function and domain of optimisation.

*Remark:* The FW design points $\{x_i^{\text{FW}}\}_{i=1}^n$ are generally not available in closed-form. We follow mainstream literature by selecting, at each iteration, the point that minimises the MMD over a finite collection of $M$ points, drawn i.i.d from $p(x)$. The authors in [18] proved that this approximation adds a $\mathcal{O}(M^{-1/4})$ term to the MMD, so that theoretical results on FW convergence continue to apply provided that $M(n) \to \infty$ sufficiently quickly. Appendix A provides full details. In practice, one may also make use of a numerical optimisation scheme in order to select the points.

## 3   A Hybrid Approach: Frank-Wolfe Bayesian Quadrature

To combine the advantages of a probabilistic integrator with a formal convergence theory, we propose Frank-Wolfe Bayesian Quadrature (FWBQ). In FWBQ, we first select design points $\{x_i^{\text{FW}}\}_{i=1}^n$ using the FW algorithm. However, when computing the quadrature approximation, instead of using the usual FW weights $\{w_i^{\text{FW}}\}_{i=1}^n$ we use instead the weights $\{w_i^{\text{BQ}}\}_{i=1}^n$ provided by BQ. We denote this quadrature rule by $\hat{p}_{\text{FWBQ}}$ and also consider $\hat{p}_{\text{FWLSBQ}}$, which uses FWLS in place of FW. As we show below, these hybrid estimators (i) carry the Bayesian interpretation of Sec. 2.2, (ii) permit a rigorous theoretical analysis, and (iii) out-perform existing FW quadrature rules by orders of magnitude in simulations. FWBQ is hence ideally suited to probabilistic numerics applications.

For these theoretical results we assume that $f$ belongs to a finite-dimensional RKHS $\mathcal{H}$, in line with recent literature [2, 10, 17, 18]. We further assume that $\mathcal{X}$ is a compact subset of $\mathbb{R}^d$, that $p(x) > 0$ $\forall x \in \mathcal{X}$ and that $k$ is continuous on $\mathcal{X} \times \mathcal{X}$. Under these hypotheses, Theorem 1 establishes consistency of the posterior mean, while Theorem 2 establishes contraction for the posterior distribution.

**Theorem 1** (Consistency). *The posterior mean $\hat{p}_{\text{FWBQ}}[f]$ converges to the true integral $p[f]$ at the following rates:*

$$\left| p[f] - \hat{p}_{\text{FWBQ}}[f] \right| \leq MMD\left(\{x_i, w_i\}_{i=1}^n\right) \leq \begin{cases} \frac{2D^2}{R} n^{-1} & \text{for FWBQ} \\ \sqrt{2}D \exp(-\frac{R^2}{2D^2}n) & \text{for FWLSBQ} \end{cases} \quad (9)$$

*where the FWBQ uses step-size $\rho_i = 1/(i+1)$, $D \in (0, \infty)$ is the diameter of the marginal polytope $\mathcal{M}$ and $R \in (0, \infty)$ gives the radius of the smallest ball of center $\mu_p$ included in $\mathcal{M}$.*

Note that all the proofs of this paper can be found in Appendix B. An immediate corollary of Theorem 1 is that FWLSBQ has an asymptotic error which is exponential in $n$ and is therefore superior to that of any QMC estimator [7]. This is not a contradiction - recall that QMC restricts attention to uniform weights, while FWLSBQ is able to propose arbitrary weightings. In addition we highlight a robustness property: Even when the assumptions of this section do not hold, one still obtains atleast a rate $\mathcal{O}_P(n^{-1/2})$ for the posterior mean using either FWBQ or FWLSBQ [8].

*Remark*: The choice of kernel affects the convergence of the FWBQ method [15]. Clearly, we expect faster convergence if the function we are integrating is 'close' to the space of functions induced by our kernel. Indeed, the kernel specifies the geometry of the marginal polytope $\mathcal{M}$, that in turn directly influences the rate constant $R$ and $D$ associated with FW convex optimisation.

Consistency is only a stepping stone towards our main contribution which establishes posterior contraction rates for FWBQ. Posterior contraction is important as these results justify, for the first time, the probabilistic numerics approach to integration; that is, we show that the *full* posterior distribution is a sensible quantification (at least asymptotically) of numerical error in the integration routine:

**Theorem 2** (Contraction). *Let $S \subseteq \mathbb{R}$ be an open neighbourhood of the true integral $p[f]$ and let $\gamma = \inf_{r \in S^c} |r - p[f]| > 0$. Then the posterior probability mass on $S^c = \mathbb{R} \setminus S$ vanishes at a rate:*

$$\text{prob}(S^c) \leq \begin{cases} \frac{2\sqrt{2}D^2}{\sqrt{\pi}R\gamma} n^{-1} \exp\left(-\frac{\gamma^2 R^2}{8D^4} n^2\right) & \text{for FWBQ} \\ \frac{2D}{\sqrt{\pi}\gamma} \exp\left(-\frac{R^2}{2D^2}n - \frac{\gamma^2}{2\sqrt{2}D}\exp\left(\frac{R^2}{2D^2}n\right)\right) & \text{for FWLSBQ} \end{cases} \quad (10)$$

*where the FWBQ uses step-size $\rho_i = 1/(i+1)$, $D \in (0, \infty)$ is the diameter of the marginal polytope $\mathcal{M}$ and $R \in (0, \infty)$ gives the radius of the smallest ball of center $\mu_p$ included in $\mathcal{M}$.*

The contraction rates are exponential for FWBQ and super-exponential for FWLBQ, and thus the two algorithms enjoy both a probabilistic interpretation and rigorous theoretical guarantees. A notable corollary is that OBQ enjoys the same rates as FWLSBQ, resolving a conjecture by Tony O'Hagan that OBQ converges exponentially [personal communication]:

**Corollary.** *The consistency and contraction rates obtained for FWLSBQ apply also to OBQ.*

## 4 Experimental Results

### 4.1 Simulation Study

To facilitate the experiments in this paper we followed [1, 2, 11, 18] and employed an exponentiated-quadratic (EQ) kernel $k(x, x') := \lambda^2 \exp(-1/2\sigma^2 \|x - x'\|_2^2)$. This corresponds to an infinite-dimensional RKHS, not covered by our theory; nevertheless, we note that all simulations are practically finite-dimensional due to rounding at machine precision. See Appendix E for a finite-dimensional approximation using random Fourier features. EQ kernels are popular in the BQ literature as, when $p$ is a mixture of Gaussians, the mean element $\mu_p$ is analytically tractable (see Appendix C). Some other $(p, k)$ pairs that produce analytic mean elements are discussed in [1].

For this simulation study, we took $p(x)$ to be a 20-component mixture of 2D-Gaussian distributions. Monte Carlo (MC) is often used for such distributions but has a slow convergence rate in $\mathcal{O}_P(n^{-1/2})$. FW and FWLS are known to converge more quickly and are in this sense preferable to MC [2]. In our simulations (Fig. 2, left), both our novel methods FWBQ and FWLSBQ decreased the MMD much faster than the FW/FWLS methods of [2]. Here, the same kernel hyper-parameters $(\lambda, \sigma) = (1, 0.8)$ were employed for all methods to have a fair comparison. This suggests that the best quadrature rules correspond to elements *outside* the convex hull of $\{\Phi(x_i)\}_{i=1}^n$. Examples of those, including BQ, often assign negative weights to features (Fig. S1 right, Appendix D).

The principle advantage of our proposed methods is that they reconcile theoretical tractability with a fully probabilistic interpretation. For illustration, Fig. 2 (right) plots the posterior uncertainty due to numerical error for a typical integration problem based on this $p(x)$. In-depth empirical studies of such posteriors exist already in the literature and the reader is referred to [3, 13, 22] for details.

Beyond these theoretically tractable integrators, SBQ seems to give even better performance as $n$ increases. An intuitive explanation is that SBQ picks $\{x_i\}_{i=1}^n$ to minimise the MMD whereas

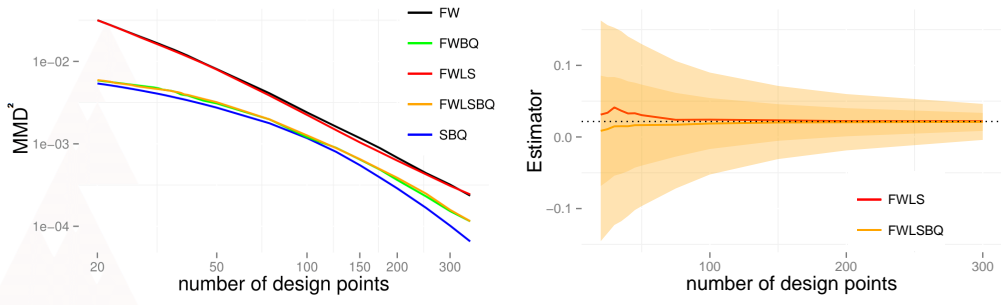

Figure 2: Simulation study. *Left*: Plot of the worst-case integration error squared (MMD$^2$). Both FWBQ and FWLSBQ are seen to outperform FW and FWLS, with SBQ performing best overall. *Right*: Integral estimates for FWLS and FWLSBQ for a function $f \in \mathcal{H}$. FWLS converges more slowly and provides only a point estimate for a given number of design points. In contrast, FWLSBQ converges faster and provides a full probability distribution over numerical error shown shaded in orange (68% and 95% credible intervals). Ground truth corresponds to the dotted black line.

FWBQ and FWLSBQ only minimise an approximation of the MMD (its linearisation along $DJ$). In addition, the SBQ weights are optimal at each iteration, which is not true for FWBQ and FWLSBQ. We conjecture that Theorem 1 and 2 provide upper bounds on the rates of SBQ. This conjecture is partly supported by Fig. 1 (right), which shows that SBQ selects similar design points to FW/FWLS (but weights them optimally). Note also that both FWBQ and FWLSBQ give very similar result. This is not surprising as FWLS has no guarantees over FW in infinite-dimensional RKHS [17].

## 4.2 Quantifying Numerical Error in a Proteomic Model Selection Problem

A topical bioinformatics application that extends recent work by [19] is presented. The objective is to select among a set of candidate models $\{M_i\}_{i=1}^m$ for protein regulation. This choice is based on a dataset $\mathcal{D}$ of protein expression levels, in order to determine a 'most plausible' biological hypothesis for further experimental investigation. Each $M_i$ is specified by a vector of kinetic parameters $\theta_i$ (full details in Appendix D). Bayesian model selection requires that these parameters are integrated out against a prior $p(\theta_i)$ to obtain marginal likelihood terms $L(M_i) = \int p(\mathcal{D}|\theta_i)p(\theta_i)\mathrm{d}\theta_i$. Our focus here is on obtaining the *maximum a posteriori* (MAP) model $M_j$, defined as the maximiser of the posterior model probability $L(M_j)/\sum_{i=1}^m L(M_i)$ (where we have assumed a uniform prior over model space). Numerical error in the computation of each term $L(M_i)$, if unaccounted for, could cause us to return a model $M_k$ that is different from the true MAP estimate $M_j$ and lead to the mis-allocation of valuable experimental resources.

The problem is quickly exaggerated when the number $m$ of models increases, as there are more opportunities for one of the $L(M_i)$ terms to be 'too large' due to numerical error. In [19], the number $m$ of models was combinatorial in the number of protein kinases measured in a high-throughput assay (currently $\sim 10^2$ but in principle up to $\sim 10^4$). This led [19] to deploy substantial computing resources to ensure that numerical error in each estimate of $L(M_i)$ was individually controlled. Probabilistic numerics provides a more elegant and efficient solution: At any given stage, we have a fully probabilistic quantification of our uncertainty in each of the integrals $L(M_i)$, shown to be sensible both theoretically and empirically. This induces a full posterior distribution over numerical uncertainty in the location of the MAP estimate (i.e. 'Bayes all the way down'). As such we can determine, on-line, the precise point in the computational pipeline when numerical uncertainty near the MAP estimate becomes acceptably small, and cease further computation.

The FWBQ methodology was applied to one of the model selection tasks in [19]. In Fig. 3 (left) we display posterior model probabilities for each of $m = 352$ candidates models, where a low number ($n = 10$) of samples were used for each integral. (For display clarity only the first 50 models are shown.) In this low-$n$ regime, numerical error introduces a second level of uncertainty that we quantify by combining the FWBQ error models for all integrals in the computational pipeline; this is summarised by a box plot (rather than a single point) for each of the models (obtained by sampling - details in Appendix D). These box plots reveal that our estimated posterior model probabilities are

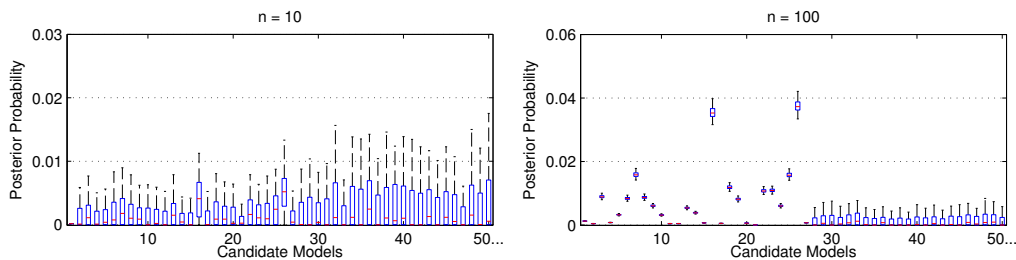

Figure 3: Quantifying numerical error in a model selection problem. FWBQ was used to model the numerical error of each integral $L(M_i)$ explicitly. For integration based on $n = 10$ design points, FWBQ tells us that the computational estimate of the model posterior will be dominated by numerical error (left). When instead $n = 100$ design points are used (right), uncertainty due to numerical error becomes much smaller (but not yet small enough to determine the MAP estimate).

completely dominated by numerical error. In contrast, when $n$ is increased through 50, 100 and 200 (Fig. 3, right and Fig. S2), the uncertainty due to numerical error becomes negligible. At $n = 200$ we can conclude that model 26 is the true MAP estimate and further computations can be halted. Correctness of this result was confirmed using the more computationally intensive methods in [19].

In Appendix D we compared the relative performance of FWBQ, FWLSBQ and SBQ on this problem. Fig. S1 shows that the BQ weights reduced the MMD by orders of magnitude relative to FW and FWLS and that SBQ converged more quickly than both FWBQ and FWLSBQ.

## 5  Conclusions

This paper provides the first theoretical results for probabilistic integration, in the form of posterior contraction rates for FWBQ and FWLSBQ. This is an important step in the probabilistic numerics research programme [15] as it establishes a theoretical justification for using the posterior distribution as a model for the numerical integration error (which was previously assumed [e.g. 11, 12, 20, 23, 25]). The practical advantages conferred by a fully probabilistic error model were demonstrated on a model selection problem from proteomics, where sensitivity of an evaluation of the MAP estimate was modelled in terms of the error arising from repeated numerical integration.

The strengths and weaknesses of BQ (notably, including scalability in the dimension $d$ of $\mathcal{X}$) are well-known and are inherited by our FWBQ methodology. We do not review these here but refer the reader to [22] for an extended discussion. Convergence, in the classical sense, was proven here to occur exponentially quickly for FWLSBQ, which partially explains the excellent performance of BQ and related methods seen in applications [12, 23], as well as resolving an open conjecture. As a bonus, the hybrid quadrature rules that we developed turned out to converge much faster in simulations than those in [2], which originally motivated our work.

A key open problem for kernel methods in probabilistic numerics is to establish protocols for the practical elicitation of kernel hyper-parameters. This is important as hyper-parameters directly affect the scale of the posterior over numerical error that we ultimately aim to interpret. Note that this problem applies equally to BQ, as well as related quadrature methods [2, 11, 12, 20] and more generally in probabilistic numerics [26]. Previous work, such as [13], optimised hyper-parameters on a per-application basis. Our ongoing research seeks automatic and general methods for hyper-parameter elicitation that provide good frequentist coverage properties for posterior credible intervals, but we reserve the details for a future publication.

### Acknowledgments

The authors are grateful for discussions with Simon Lacoste-Julien, Simo Särkkä, Arno Solin, Dino Sejdinovic, Tom Gunter and Mathias Cronjäger. FXB was supported by EPSRC [EP/L016710/1]. CJO was supported by EPSRC [EP/D002060/1]. MG was supported by EPSRC [EP/J016934/1], an EPSRC Established Career Fellowship, the EU grant [EU/259348] and a Royal Society Wolfson Research Merit Award.

## Footnotes

[1] A detailed discussion on probabilistic numerics and an extensive up-to-date bibliography can be found at `http://www.probabilistic-numerics.org`.

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
