[Supplementary Material]

## Supplementary Material

### Appendix A: Details for the FWBQ and FWLSBQ Algorithms

A high-level pseudo-code description for the Frank-Wolfe Bayesian Quadrature (FWBQ) algorithm is provided below.

---

**Algorithm 1** The Frank-Wolfe Bayesian Quadrature (FWBQ) Algorithm

---

**Require:** function $f$, reproducing kernel $k$, initial point $x_0 \in \mathcal{X}$.
1: Compute design points $\{x_i^{\text{FW}}\}_{i=1}^n$ using the FW algorithm (Alg. 1).
2: Compute associated weights $\{w_i^{\text{BQ}}\}_{i=1}^n$ using BQ (Eqn. 4).
3: Compute the posterior mean $\hat{p}_{\text{FWBQ}}[f]$, i.e. the quadrature rule with $\{x_i^{\text{FW}}, w_i^{\text{BQ}}\}_{i=1}^n$.
4: Compute the posterior variance $v_{\text{BQ}}(\{x_i^{\text{FW}}\}_{i=1}^n)$ using BQ (Eqn. 5).
5: Return the full posterior $\mathcal{N}(\hat{p}_{\text{FWBQ}}, v_{\text{BQ}}(\{x_i^{\text{FW}}\}_{i=1}^n))$ for the integral $p[f]$.

---

Frank-Wolfe Line-Search Bayesian Quadrature (FWLSBQ) is simply obtained by substituting the Frank-Wolfe algorithm with the Frank-Wolfe Line-Search algorithm. In this appendix, we derive all of the expressions necessary to implement both the FW and FWLS algorithms (for quadrature) in practice. All of the other steps can be derived from the relevant equations as highlighted in Algorithm 1 above.

The FW/FWLS are both initialised by the user choosing a design point $x_1^{\text{FW}}$. This can be done either at random or by choosing a location which is known to have high probability mass under $p(x)$. The first approximation to $\mu_p$ is therefore given by $g_1 = k(\cdot, x_1^{\text{FW}})$. The algorithm then loops over the next three steps to obtain new design points $\{x_i^{\text{FW}}\}_{i=2}^n$:

*Step 1) Obtaining the new Frank-Wolfe design points $x_{i+1}^{FW}$.*

At iteration $i$, the step consists of choosing the point $\bar{x}_i^{\text{FW}}$. Let $\{w_l^{(i)}\}_{l=1}^{i-1}$ denote the Frank-Wolfe weights assigned to each of the previous design points $\{x_l^{\text{FW}}\}_{l=1}^{i-1}$ at this new iteration, given that we choose $x$ as our new design point. The choice of new design point is done by computing the derivative of the objective function $J(g_{i-1})$ and finding the point $x^*$ which minimises the inner product:

$$\arg\min_{g \in \mathcal{G}} \langle g, (DJ)(g_{i-1}) \rangle_\times \tag{1}$$

To do so, we need to obtain an equivalent expression of the minimisation of the linearisation of $J$ (denoted $DJ$) in terms of kernel values and evaluations of the mean element $\mu_p$. Since minimisation of a linear function can be restricted to extreme points of the domain, we have that

$$\arg\min_{g \in \mathcal{G}} \langle g, (DJ)(g_{i-1}) \rangle_\times = \arg\min_{x \in \mathcal{X}} \langle \Phi(x), (DJ)(g_{i-1}) \rangle_\times. \tag{2}$$

Then using the definition of $J$ we have:

$$\arg\min_{x \in \mathcal{X}} \langle \Phi(x), (DJ)(g_{i-1}) \rangle_\times = \arg\min_{x \in \mathcal{X}} \langle \Phi(x), g_{i-1} - \mu_p \rangle_\mathcal{H}, \tag{3}$$

where

$$
\begin{aligned}
\langle \Phi(x), g_{i-1} - \mu_p \rangle_\mathcal{H} &= \left\langle \Phi(x), \sum_{l=1}^{i-1} w_l^{(i-1)} \Phi(x_l) - \mu_p \right\rangle_\mathcal{H} \\
&= \sum_{i=1}^{i-1} w_l^{(i-1)} \langle \Phi(x), \Phi(x_l) \rangle_\mathcal{H} - \langle \Phi(x), \mu_p \rangle_\mathcal{H} \\
&= \sum_{l=1}^{i-1} w_l^{(i-1)} k(x, x_l) - \mu_p(x).
\end{aligned}
\tag{4}
$$

Our new design point $x_i^{\text{FW}}$ is therefore the point $x^*$ which minimises this expression. Note that this equation may not be convex and may require us to make use of approximate methods to find the

minimum $x^*$. To do so, we sample $M$ points (where $M$ is large) independently from the distribution $p$ and pick the sample which minimises the expression above. From [5] this introduces an additive error term of size $\mathcal{O}(M^{-1/4})$, which does not impact our convergence analysis provided that M(n) vanishes sufficiently quickly. In all experiments we took $M$ between $10,000$ and $50,000$ so that this error will be negligible.

It is important to note that sampling from $p(x)$ is likely to not be the best solution to optimising this expression. One may, for example, be better off using any other optimisation method which does not require convexity (for example, Bayesian Optimization). However, we have used sampling as the result from [5] discussed above allows us to have a theoretical upper bound on the error introduced.

*Step 2) Computing the Step-Sizes and Weights for the Frank-Wolfe and Frank-Wolfe Line-Search Algorithms.*

Computing the weights $\{w_l^{(i)}\}_{l=1}^n$ assigned by the FW/FWLS algorithms to each of the design points is obtained using the equation:

$$w_l^{(i)} = \prod_{j=l+1}^{i} \left(1 - \rho_{j-1}\right)\rho_{l-1} \tag{5}$$

Clearly, this expression depends on the choice of step-sizes $\{\rho_l\}_{l=1}^i$. In the case of the standard Frank-Wolfe algorithm, this step-size sequence is a an input from the algorithm and so computing the weights is straightforward. However, in the case of the Frank-Wolfe Line-Search algorithm, the choice of step-size is optimized at each iteration so that $g_i$ minimises $J$ the most.

In the case of computing integrals, this optimization step can actually be obtained analytically. This analytic expression will be given in terms of values of the kernel values and evaluations of the mean element.

First, from the definition of $J$

$$
\begin{aligned}
J\big((1-\rho)g_{i-1} + \rho\Phi(x_i)\big) &= \frac{1}{2}\big\langle (1-\rho)g_{i-1} + \rho\Phi(x_i) - \mu_p, (1-\rho)g_{i-1} + \rho\Phi(x_i) - \mu_p\big\rangle_{\mathcal{H}} \\
&= \frac{1}{2}\Big[(1-\rho)^2\big\langle g_{i-1}, g_{i-1}\big\rangle_{\mathcal{H}} + 2(1-\rho)\rho\big\langle g_{i-1}, \Phi(x_i)\big\rangle_{\mathcal{H}} \\
&\quad + 2\rho^2\big\langle \Phi(x_i), \Phi(x_i)\big\rangle_{\mathcal{H}} - 2(1-\rho)\big\langle g_{i-1}, \mu_p\big\rangle_{\mathcal{H}} \\
&\quad - 2\rho\big\langle \Phi(x_i), \mu_p\big\rangle_{\mathcal{H}} + \big\langle \mu_p, \mu_p\big\rangle_{\mathcal{H}}\Big].
\end{aligned}
\tag{6}
$$

Taking the derivative of this expression with respect to $\rho$, we get:

$$
\begin{aligned}
\frac{\partial J\big((1-\rho)g_{i-1} + \rho\Phi(x_i)\big)}{\partial \rho} &= \frac{1}{2}\Big[-2(1-\rho)\big\langle g_{i-1}, g_{i-1}\big\rangle_{\mathcal{H}} + 2(1-2\rho)\big\langle g_{i-1}, \Phi(x_i)\big\rangle_{\mathcal{H}} \\
&\quad + 2\rho\big\langle \Phi(x_i), \Phi(x_i)\big\rangle_{\mathcal{H}} + 2\big\langle g_{i-1}, \mu_p\big\rangle_{\mathcal{H}} - 2\big\langle \Phi(x_i), \mu_p\big\rangle_{\mathcal{H}}\Big] \\
&= \rho\Big[\big\langle g_{i-1}, g_{i-1}\big\rangle_{\mathcal{H}} - 2\big\langle g_{i-1}, \Phi(x_i)\big\rangle_{\mathcal{H}} + \big\langle \Phi(x_i), \Phi(x_i)\big\rangle_{\mathcal{H}} \\
&= \rho\big\| g_{i-1} - \Phi(x_i)\big\|_{\mathcal{H}}^2 - \big\langle g_{i-1} - \Phi(x_i), g_{i-1} - \mu_p\big\rangle_{\mathcal{H}}.
\end{aligned}
\tag{7}
$$

Setting this derivative to zero gives us the following optimum:

$$
\rho^* = \frac{\big\langle g_{i-1} - \mu_p, g_{i-1} - \Phi(x_i)\big\rangle_{\mathcal{H}}}{\big\| g_{i-1} - \Phi(x_i)\big\|_{\mathcal{H}}^2}.
\tag{8}
$$

Clearly, differentiating a second time with respect to $\rho$ gives $\|g_{i-1} - \Phi(x_i)\|_{\mathcal{H}}^2$, which is non-negative and so $\rho^*$ is a minimum. One can show using geometrical arguments about the marginal polytope $\mathcal{M}$ that $\rho^*$ will be in $[0, 1]$ [4].

The numerator of this line-search expression is

$$
\begin{aligned}
\left\langle g_{i-1} - \mu_p, g_{i-1} - \Phi(x_i) \right\rangle_{\mathcal{H}} &= \left\langle g_{i-1}, g_{i-1} \right\rangle_{\mathcal{H}} - \left\langle \mu_p, g_{i-1} \right\rangle_{\mathcal{H}} - \sum_{l=1}^{i-1} w_l^{(i-1)} k(x_l, x_i) + \mu_p(x_i) \\
&= \sum_{l=1}^{i-1}\sum_{m=1}^{i-1} w_l^{(i-1)} w_m^{(i-1)} k(x_l, x_m) - \sum_{l=1}^{i-1} w_l^{(i-1)}\Big[ k(x_l, x_i) + \mu_p(x_l) \Big] \\
&\quad + \mu_p(x_i).
\end{aligned}
\tag{9}
$$

Similarly the denominator is

$$
\begin{aligned}
\left\| g_{i-1} - \Phi(x_i) \right\|_{\mathcal{H}}^2 &= \left\langle g_{i-1} - \Phi(x_i), g_{i-1} - \Phi(x_i) \right\rangle_{\mathcal{H}} \\
&= \left\langle g_{i-1}, g_{i-1} \right\rangle_{\mathcal{H}} - 2\left\langle g_{i-1}, \Phi(x_i) \right\rangle_{\mathcal{H}} + \left\langle \Phi(x_i), \Phi(x_i) \right\rangle_{\mathcal{H}} \\
&= \sum_{l=1}^{i-1}\sum_{m=1}^{i-1} w_l^{(i-1)} w_m^{(i-1)} k(x_l, x_m) - 2\sum_{l=1}^{i-1} w_l^{(i-1)} k(x_l, x_i) + k(x_i, x_i).
\end{aligned}
\tag{10}
$$

Clearly all expressions provided here can be vectorised for efficient computational implementation.

*Step 3) Computing a new approximation of the mean element.*

The final step consists of updating the approximation of the mean element, which can be done directly by setting:

$$
g_i = (1 - \rho_i) g_{i-1} + \rho_i \bar{g}_i
\tag{11}
$$

## Appendix B: Proofs of Theorems and Corollaries

**Theorem** (Consistency). *The posterior mean $\hat{p}_{\mathrm{FWBQ}}[f]$ converges to the true integral $p[f]$ at the following rates:*

$$
\left| p[f] - \hat{p}_{\mathrm{FWBQ}}[f] \right| \le MMD\big(\{x_i, w_i\}_{i=1}^n\big) \le
\begin{cases}
\frac{2D^2}{R} n^{-1} & \text{for FWBQ} \\
\sqrt{2} D \exp(-\frac{R^2}{2D^2} n) & \text{for FWLSBQ}
\end{cases}
$$

*where the FWBQ uses step-size $\rho_i = 1/(i+1)$, $D \in (0, \infty)$ is the diameter of the marginal polytope $\mathcal{M}$ and $R \in (0, \infty)$ gives the radius of the smallest ball of center $\mu_p$ included in $\mathcal{M}$.*

*Proof.* The posterior mean in BQ is a Bayes estimator and so the MMD takes a minimax form [3]. In particular, the BQ weights perform no worse than the FW weights:

$$
\mathrm{MMD}\Big(\{x_i^{\mathrm{FW}}, w_i^{\mathrm{BQ}}\}_{i=1}^n\Big) = \inf_{\mathbf{w} \in \mathbb{R}^n} \mathrm{MMD}\Big(\{x_i^{\mathrm{FW}}, w_i\}_{i=1}^n\Big) \le \mathrm{MMD}\Big(\{x_i^{\mathrm{FW}}, w_i^{\mathrm{FW}}\}_{i=1}^n\Big).
\tag{12}
$$

Now, the values attained by the objective function $J$ along the path $\{g_i\}_{i=1}^n$ determined by the FW(/FWLS) algorithm can be expressed in terms of the MMD as follows:

$$
J(g_n) = \frac{1}{2}\left\| \hat{\mu}_{\mathrm{FW}} - \mu_p \right\|_{\mathcal{H}}^2 = \frac{1}{2}\mathrm{MMD}^2\Big(\{x_i^{\mathrm{FW}}, w_i^{\mathrm{FW}}\}_{i=1}^n\Big).
\tag{13}
$$

Combining (12) and (13) gives

$$
\left| p[f] - \hat{p}_{\mathrm{FWBQ}}[f] \right| \le \mathrm{MMD}\Big(\{x_i^{\mathrm{FW}}, w_i^{\mathrm{BQ}}\}_{i=1}^n\Big) \|f\|_{\mathcal{H}} \le 2^{1/2} J^{1/2}(g_n),
\tag{14}
$$

since $\|f\|_{\mathcal{H}} \le 1$. To complete the proof we leverage recent analysis of the FW algorithm with steps $\rho_i = 1/(n+1)$ and the FWLS algorithm. Specifically, from [2, Prop. 1] we have that:

$$
J(g_n) \le
\begin{cases}
\frac{2D^4}{R^2} n^{-2} & \text{for FW with step size } \rho_i = 1/(i+1) \\
D^2 \exp(-R^2 n / D^2) & \text{for FWLS}
\end{cases}
\tag{15}
$$

where $D$ is the diameter of the marginal polytope $\mathcal{M}$ and $R$ is the radius of the smallest ball centered at $\mu_p$ included in $\mathcal{M}$. $\qquad\square$

**Theorem** (Contraction). *Let $S \subseteq \mathbb{R}$ be an open neighbourhood of the true integral $p[f]$ and let $\gamma = \inf_{r \in S^c} |r - p[f]| > 0$. Then the posterior probability mass on $S^c = \mathbb{R} \setminus S$ vanishes at a rate:*

$$\text{prob}(S^c) \leq \begin{cases} \frac{2\sqrt{2}D^2}{\sqrt{\pi}R\gamma} n^{-1} \exp\left(-\frac{\gamma^2 R^2}{8D^4} n^2\right) & \text{for FWBQ, } \rho_i = 1/(i+1) \\ \frac{2D}{\sqrt{\pi}\gamma} \exp\left(-\frac{R^2}{2D^2}n - \frac{\gamma^2}{2\sqrt{2}D} \exp\left(\frac{R^2}{2D^2}n\right)\right) & \text{for FWLSBQ} \end{cases}$$

*where $D \in (0, \infty)$ is the diameter of the marginal polytope $\mathcal{M}$ and $R \in (0, \infty)$ gives the radius of the smallest ball of center $\mu_p$ included in $\mathcal{M}$.*

*Proof.* We will obtain the posterior contraction rates of interest using the bounds on the MMD provided in the proof of Theorem 1. Given an open neighbourhood $S \subseteq \mathbb{R}$ of $p[f]$, we have that the complement $S^c = \mathbb{R} \setminus S$ is closed in $\mathbb{R}$. We assume without loss of generality that $S^c \neq \emptyset$, since the posterior mass on $S^c$ is trivially zero when $S^c = \emptyset$. Since $S^c$ is closed, the distance $\gamma = \inf_{r \in S^c} |r - p[f]| > 0$ is strictly positive. Denote the posterior distribution by $\mathcal{N}(m_n, \sigma_n^2)$ where we have that $m_n := \hat{p}_{\text{FWBQ}}[f]$ where $\hat{p}_{\text{FWBQ}} = \sum_{i=1}^n w_i^{\text{BQ}} \delta(x_i^{\text{FW}})$ and $\sigma_n := \text{MMD}(\{x_i^{\text{FW}}, w_i^{\text{BQ}}\}_{i=1}^n)$. Directly from the supremum definition of the MMD we have:

$$\left|p[f] - m_n\right| \leq \sigma_n \|f\|_{\mathcal{H}}. \tag{16}$$

Now the posterior probability mass on $S^c$ is given by

$$M_n = \int_{S^c} \phi(r|m_n, \sigma_n) \mathrm{d}r, \tag{17}$$

where $\phi(r|m_n, \sigma_n)$ is the p.d.f. of the posterior normal distribution. By the definition of $\gamma$ we get the upper bound:

$$M_n \leq \int_{-\infty}^{p[f]-\gamma} \phi(r|m_n, \sigma_n)\mathrm{d}r + \int_{p[f]+\gamma}^{\infty} \phi(r|m_n, \sigma_n)\mathrm{d}r \tag{18}$$

$$= 1 + \Phi\left(\underbrace{\frac{p[f]-m_n}{\sigma_n}}_{(*)} - \frac{\gamma}{\sigma_n}\right) - \Phi\left(\underbrace{\frac{p[f]-m_n}{\sigma_n}}_{(*)} + \frac{\gamma}{\sigma_n}\right). \tag{19}$$

From (16) we have that the terms $(*)$ are bounded by $\|f\|_{\mathcal{H}} \leq 1 < \infty$ as $\sigma_n \to 0$, so that asymptotically we have:

$$M_n \lesssim 1 + \Phi\left(-\gamma/\sigma_n\right) - \Phi\left(\gamma/\sigma_n\right) \tag{20}$$

$$= \text{erfc}\left(\gamma/\sqrt{2}\sigma_n\right) \sim \left(\sqrt{2}\sigma_n/\sqrt{\pi}\gamma\right) \exp\left(-\gamma^2/2\sigma_n^2\right). \tag{21}$$

Finally we may substitute the asymptotic results derived in the proof of Theorem 1 for the MMD $\sigma_n$ into (21) to complete the proof. $\qquad\square$

**Corollary.** *The consistency and contraction rates obtained for FWLSBQ apply also to OBQ.*

*Proof.* By definition, OBQ chooses samples that globally minimise the MMD and we can hence bound this quantity from above by the MMD of FWLSBQ:

$$\text{MMD}\left(\{x_i^{\text{OBQ}}, w_i^{\text{BQ}}\}_{i=1}^n\right) = \inf_{\{x_i\}_{i=1}^n \in \mathcal{X}} \text{MMD}\left(\{x_i, w_i^{\text{BQ}}\}_{i=1}^n\right) \leq \text{MMD}\left(\{x_i^{\text{FW}}, w_i^{\text{BQ}}\}_{i=1}^n\right). \tag{22}$$

Consistency and contraction follow from inserting this inequality into the above proofs. $\qquad\square$

**Appendix C: Computing the Mean Element for the Simulation Study**

We compute an expression for $\mu_p(x) = \int_{-\infty}^{\infty} k(x, x')p(x')\mathrm{d}x'$ in the case where $k$ is an exponentiated-quadratic kernel with length scale hyper-parameter $\sigma$:

$$k(x, x') := \lambda^2 \exp\left(\frac{-\sum_{i=1}^d (x_i - x_i')^2}{2\sigma^2}\right) = \lambda^2 (\sqrt{2\pi}\sigma)^d \phi(x|x', \Sigma_\sigma), \tag{23}$$

where $\Sigma_\sigma$ is a d-dimensional diagonal matrix with entries $\sigma^2$, and where $p(x)$ is a mixture of d-dimensional Gaussian distributions:

$$p(x) \quad = \quad \sum_{l=1}^{L} \rho_l \, \phi\big(x\big|\mu_l, \Sigma_l\big). \tag{24}$$

(Note that, in this section only, $x_i$ denotes the $i$th component of the vector $x$.) Using properties of Gaussian distributions (see Appendix A.2 of [8]) we obtain

$$
\begin{aligned}
\mu_p(x) \quad &= \quad \int_{-\infty}^{\infty} k(x, x') p(x') \mathrm{d}x' \\
&= \quad \int_{-\infty}^{\infty} \lambda^2 (\sqrt{2\pi}\sigma)^d \phi\big(x'\big|x, \Sigma_\sigma\big) \times \Big( \sum_{l=1}^{L} \rho_l \, \phi\big(x'\big|\mu_l, \Sigma_l\big) \Big) \mathrm{d}x' \\
&= \quad \lambda^2 (\sqrt{2\pi}\sigma)^d \sum_{l=1}^{L} \rho_l \int_{-\infty}^{\infty} \phi\big(x'\big|x, \Sigma_\sigma\big) \times \phi\big(x'\big|\mu_l, \Sigma_l\big) \mathrm{d}x' \\
&= \quad \lambda^2 (\sqrt{2\pi}\sigma)^d \sum_{l=1}^{L} \rho_l \int_{-\infty}^{\infty} a_l^{-1} \phi\big(x'\big|c_l, C_l\big) \mathrm{d}x' \\
&= \quad \lambda^2 (\sqrt{2\pi}\sigma)^d \sum_{l=1}^{L} \rho_l a_l^{-1}.
\end{aligned}
\tag{25}
$$

where we have:

$$a_l^{-1} \quad = \quad (2\pi)^{-\frac{d}{2}} \big|\Sigma_\sigma + \Sigma_l\big|^{-\frac{1}{2}} \exp\big( -\frac{1}{2}(x - \mu_l)^T (\Sigma_\sigma + \Sigma_l)^{-1} (x - \mu_l)\big). \tag{26}$$

This last expression is in fact itself a Gaussian distribution with probability density function $\phi(x|\mu_l, \Sigma_l + \Sigma_\sigma)$ and we hence obtain:

$$\mu_p(x) \quad := \quad \lambda^2 \big(\sqrt{2\pi}\sigma\big)^d \sum_{l=1}^{L} \rho_l \, \phi\big(x|\mu_l, \Sigma_l + \Sigma_\sigma\big). \tag{27}$$

Finally, we once again use properties of Gaussians to obtain

$$
\begin{aligned}
\int_{-\infty}^{\infty} \mu_p(x) p(x) \mathrm{d}x \quad &= \quad \int_{-\infty}^{\infty} \Big[ \lambda^2 \big(\sqrt{2\pi}\sigma\big)^d \sum_{l=1}^{L} \rho_l \, \phi\big(x|\mu_l, \Sigma_l + \Sigma_\sigma\big) \Big] \times \Big[ \sum_{m=1}^{L} \rho_m \, \phi\big(x|\mu_m, \Sigma_m\big) \Big] \mathrm{d}x \\
&= \quad \lambda^2 \big(\sqrt{2\pi}\sigma\big)^d \sum_{l=1}^{L} \sum_{m=1}^{L} \rho_l \rho_m \int_{-\infty}^{\infty} \phi\big(x|\mu_l, \Sigma_l + \Sigma_\sigma\big) \phi\big(x|\mu_m, \Sigma_m\big) \mathrm{d}x \\
&= \quad \lambda^2 \big(\sqrt{2\pi}\sigma\big)^d \sum_{l=1}^{L} \sum_{m=1}^{L} \rho_l \rho_m a_{lm}^{-1} \\
&= \quad \lambda^2 \big(\sqrt{2\pi}\sigma\big)^d \sum_{l=1}^{L} \sum_{m=1}^{L} \rho_l \rho_m \phi\big(\mu_l|\mu_m, \Sigma_l + \Sigma_m + \Sigma_\sigma\big).
\end{aligned}
\tag{28}
$$

Other combinations of kernel $k$ and density $p$ that give rise to an analytic mean element can be found in the references of [1].

## Appendix D: Details of the Application to Proteomics Data

*Description of the Model Choice Problem*

The 'CheMA' methodology described in [6] contains several elements that we do not attempt to reproduce in full here; in particular we do not attempt to provide a detailed motivation for the mathematical forms presented below, as this requires elements from molecular chemistry. For our

present purposes it will be sufficient to define the statistical models $\{M_i\}_{i=1}^m$ and to clearly specify the integration problems that are to be solved. We refer the reader to [6] and the accompanying supplementary materials for a full biological background.

Denote by $\mathcal{D}$ the dataset containing normalised measured expression levels $y_S(t_j)$ and $y_S^*(t_j)$ for, respectively, the unphosphorylated and phosphorylated forms of a protein of interest ('substrate') in a longitudinal experiment at time $t_j$. In addition $\mathcal{D}$ contains normalised measured expression levels $y_{E_i}^*(t_j)$ for a set of possible regulator kinases ('enzymes', here phosphorylated proteins) that we denote by $\{E_i\}$.

An important scientific goal is to identify the roles of enzymes (or 'kinases') in protein signaling; in this case the problem takes the form of variable selection and we are interested to discover which enzymes must be included in a model for regulation of the substrate $S$. Specifically, a candidate model $M_i$ specifies which enzymes in the set $\{E_i\}$ are regulators of the substrate $S$, for example $M_3 = \{E_2, E_4\}$. Following [6] we consider models containing at most two enzymes, as well as the model containing no enzymes.

Given a dataset $\mathcal{D}$ and model $M_i$, we can write down a likelihood function as follows:

$$L(\theta_i, M_i) = \prod_{n=1}^N \phi \left( \frac{y_S^*(t_{n+1}) - y_S^*(t_n)}{t_{n+1} - t_n} \, \middle| - \frac{V_0 y_S^*(t_n)}{y_S^*(t_n) + K_0} + \sum_{E_j \in M_i} \frac{V_j y_{E_j}^*(t_n) y_S(t_n)}{y_S(t_n) + K_j}, \sigma_{\text{err}}^2 \right). \quad (29)$$

Here the model parameters are $\theta_i = \{\text{K}, \text{V}, \sigma_{\text{err}}\}$, where $(\text{K})_j = K_j$, $(\text{V})_j = V_j$, $\phi$ is the normal p.d.f. and the mathematical forms arise from the Michaelis-Menten theory of enzyme kinetics. The $V_j$ are known as 'maximum reaction rates' and the $K_j$ are known as 'Michaelis-Menten parameters'. This is classical chemical notation, not to be confused with the kernel matrix from the main text. The final parameter $\sigma_{\text{err}}$ defines the error magnitude for this 'approximate gradient-matching' statistical model.

The prior specification proposed in [6] and followed here is

$$\text{K} \quad \sim \quad \phi_T\big(K\big|1, 2^{-1}\text{I}\big), \quad (30)$$

$$\sigma_{\text{err}}|\text{K} \quad \sim \quad p(\sigma_{\text{err}}) \propto 1/\sigma_{\text{err}}, \quad (31)$$

$$\text{V}|\text{K}, \sigma \quad \sim \quad \phi_T\big(V\big|1, N\sigma_{\text{err}}^2\big(\text{X}(\text{K})^T\text{X}(\text{K})\big)^{-1}\big), \quad (32)$$

where $\phi_T$ denotes a Gaussian distribution, truncated so that its support is $[0, \infty)$ (since kinetic parameters cannot be non-negative). Here $\text{X}(\text{K})$ is the design matrix associated with the linear regression that is obtained by treating the K as known constants; we refer to [6] for further details.

Due to its careful design, the likelihood in Eqn. 29 is partially conjugate, so the following integral can be evaluated in closed form:

$$L(\text{K}, M_i) = \int_0^\infty \int_0^\infty L(\theta_i, M_i) p(\text{V}, \sigma_{\text{err}}|\text{K}) \mathrm{d}\text{V}\mathrm{d}\sigma_{\text{err}}. \quad (33)$$

The numerical challenge is then to compute the integral

$$L(M_i) = \int_0^\infty L(\text{K}, M_i) p(\text{K}) \mathrm{d}\text{K}, \quad (34)$$

for each candidate model $M_i$. Depending on the number of enzymes in model $M_i$, this will either be a 1-, 2- or 3-dimensional numerical integral. Whilst such integrals are not challenging to compute on a per-individual basis, the nature of the application means that the values $L(M_i)$ will be similar for many candidate models and, when the number of models is large, this demands either a very precise calculation per model or a careful quantification of the impact of numerical error on the subsequent inferences (i.e. determining the MAP estimate). It is this particular issue that motivates the use of probabilistic numerical methods.

*Description of the Computational Problem*

We need to compute integrals of functions with domain $\mathcal{X} = [0, \infty)^d$ where $d \in \{1, 2, 3\}$ and the sampling distribution $p(x)$ takes the form $\phi_T(x|1, 2^{-1}\text{I})$. The test function $f(x)$ corresponds to

$L(\mathrm{K}, M_i)$ with $x = \mathrm{K}$. This is given explicitly by the $g$-prior formulae as:

$$L(\mathrm{K}, M_i) \;=\; \frac{1}{(2\pi)^{N/2}} \frac{1}{(N+1)^{d/2}} \Gamma\left(\frac{N}{2}\right) b_N^{-\frac{N}{2}}, \tag{35}$$

$$b_N \;=\; \frac{1}{2}\left(\mathrm{Y}^T\mathrm{Y} + \frac{1}{N}\mathbf{1}^T\mathrm{X}^T\mathrm{X}\mathbf{1} - \mathrm{V}_N^T\Omega_N\mathrm{V}_N\right), \tag{36}$$

$$\mathrm{V}_N \;=\; \Omega_N^{-1}\left(\frac{1}{N}\mathrm{X}^T\mathrm{X}\mathbf{1} + \mathrm{X}^T\mathrm{Y}\right), \tag{37}$$

$$\Omega_N \;=\; \left(1 + \frac{1}{N}\right)\mathrm{X}^T\mathrm{X}, \tag{38}$$

$$(\mathrm{Y})_n \;=\; \frac{y_S^*(t_{n+1}) - y_S^*(t_n)}{t_{n+1} - t_n}, \tag{39}$$

$$\tag{40}$$

where for clarity we have suppressed the dependence of X on K. For the Frank-Wolfe Bayesian Quadrature algorithm, we require that the mean element $\mu_p$ is analytically tractable and for this reason we employed the exponentiated-quadratic kernel with length scale $\lambda$ and width scale $\sigma$ parameters:

$$k(x, x') = \lambda^2 \exp\left(-\frac{\sum_{i=1}^d (x_i - x_i')^2}{2\sigma^2}\right). \tag{41}$$

For simplicity we focussed on the single hyper-parameter pair $\lambda = \sigma = 1$, which produces:

$$\mu_p(x) \;=\; \int_0^\infty k(x, x')p(x')\mathrm{d}x' \tag{42}$$

$$=\; \int_0^\infty \exp\left(-\sum_{i=1}^d (x_i - x_i')^2\right)\phi_T\left(x'\big|1, 2^{-1}\mathrm{I}\right)\mathrm{d}x' \tag{43}$$

$$=\; 2^{-d/2}\left(1 + \mathrm{erf}(1)\right)^{-d}\prod_{i=1}^d \exp\left(-\frac{(x_i - 1)^2}{2}\right)\left(1 + \mathrm{erf}\left(\frac{x_i + 1}{\sqrt{2}}\right)\right), \tag{44}$$

where $\phi_T$ is the p.d.f. of the truncated Gaussian distribution introduced above and erf is the error function. To compute the posterior variance of the numerical error we also require the quantity:

$$\int_0^\infty \int_0^\infty k(x, x')p(x)p(x')\mathrm{d}x\mathrm{d}x' = \int_0^\infty \mu_p(x)p(x)\mathrm{d}x = \begin{cases} 0.629907... & \text{for } d = 1 \\ 0.396783... & \text{for } d = 2 \\ 0.249937... & \text{for } d = 3 \end{cases}, \tag{45}$$

which we have simply evaluated numerically. We emphasise that principled approaches to hyper-parameter elicitation are an important open research problem that we aim to address in a future publication (see discussion in the main text). The values used here are scientifically reasonable and serve to illustrate key aspects of our methodology.

FWBQ provides posterior distributions over the numerical uncertainty in each of our estimates for the marginal likelihoods $L(M_i)$. In order to propagate this uncertainty forward into a posterior distribution over posterior model probabilities (see Figs. 3 in the main text and S2 below), we simply sampled values $\hat{L}(M_i)$ from each of the posterior distributions for $L(M_i)$ and used these samples values to construct posterior model probabilities $\hat{L}(M_i)/\sum_j \hat{L}(M_j)$. Repeating this procedure many times enables us to sample from the posterior distribution over the posterior model probabilities (i.e. two levels of Bayes' theorem). This provides a principled quantification of the uncertainty due to numerical error in the output of our primary Bayesian analysis.

*Description of the Data*

The proteomic dataset $\mathcal{D}$ that we considered here was a subset of the larger dataset provided in [6]. Specifically, the substrate $S$ was the well-studied 4E-binding protein 1 (4EBP1) and the enzymes $E_j$ consisted of a collection of key proteins that are thought to be connected with 4EBP1 regulation, or at least involved in similar regulatory processes within cellular signalling. Full details, including

experimental protocols, data normalisation and the specific choice of measurement time points are provided in the supplementary materials associated with [6].

For this particular problem, biological interest arises because the data-generating system was provided by breast cancer cell lines. As such, the textbook description of 4EBP1 regulation may not be valid and indeed it is thought that 4EBP1 dis-regulation is a major contributing factor to these complex diseases (see [9]). We do not elaborate further on the scientific rationale for model-based proteomics in this work.

Figure S1: Comparison of quadrature methods on the proteomics dataset. *Left:* Value of the $MMD^2$ for FW (black), FWLS (red), FWBQ (green), FWLSBQ (orange) and SBQ (blue). Once again, we see the clear improvement of using Bayesian Quadrature weights and we see that Sequential Bayesian Quadrature improves on Frank-Wolfe Bayesian Quadrature and Frank-Wolfe Line-Search Bayesian Quadrature. *Right:* Empirical distribution of weights. The dotted line represent the weights of the Frank-Wolfe algorithm with line search, which has all weights $w_i = 1/n$. Note that the distribution of Bayesian Quadrature weights ranges from $-17.39$ to $13.75$ whereas all versions of Frank-Wolfe have weights limited to $[0, 1]$ and have to sum to 1.

Figure S2: Quantifying numerical error in a model selection problem. Marginalisation of model parameters necessitates numerical integration and any error in this computation will introduce error into the reported posterior distribution over models. Here FWBQ is used to model this numerical error explicitly. *Left*: At $n = 50$ design points the uncertainty due to numerical error prevents us from determining the true MAP estimate. *Right*: At $n = 200$ design points, models 16 and 26 can be better distinguished as the uncertainty due to numerical error is reduced (model 26 can be seen to be the MAP estimate, although some uncertainty about this still remains even at this value of $n$, due to numerical error).

**Appendix E: FWBQ algorithms with Random Fourier Features**

In this section, we will investigate the use of random Fourier features (introduced in [7]) for the FWLS and FWLSBQ algorithms. An advantage of using this type of approximation is that the cost of manipulating the Gram matrix, and in particular of inverting it, goes down from $\mathcal{O}(n^3)$ to $\mathcal{O}(nD^2)$ for some user-defined constant $D$ which controls the quality of approximation. This could make Bayesian Quadrature more competitive against other integration methods such as MCMC or QMC. Furthermore, the kernels obtained using this method lead to finite-dimensional RKHS, which therefore satisfy the assumptions required for the theory in this paper to hold. This will be the aspect that we will focus on. In particular, we will show empirically that exponential convergence may be possible even when the RKHS is infinite-dimensional.

We will re-use the 20-component mixture of Gaussians example with $d = 2$ from our simulation studies, but using instead a random Fourier approximation of the exponentiated-quadratic (EQ) kernel $k(x, x') := \lambda^2 \exp(-1/2\sigma^2 \|x - x'\|_2^2)$ with $(\lambda, \sigma) = (1, 0.8)$ and $M = 10000$.

Following Bochner's theorem, we can always express translation invariant kernels in Fourier space:

$$k(x, x') = \int_{\mathcal{W}} g(w) \exp\left(jw(x - x')\right) \mathrm{d}w = \mathbb{E}\left[ \exp\left(jw^T x\right) \exp\left(jw^T x'\right) \right] \tag{46}$$

where $w \sim g(w)$ for $g(w)$ being the Fourier transform of the kernel. One can then use a Monte Carlo approximation of the kernel's Fourier expression with $D$ samples whenever $g$ is a p.d.f.. Our approximated kernel will then lead to a $D$-dimensional RKHS and will be given by:

$$k(x, x') \approx \frac{1}{D} \sum_{j=1}^{D} z_{w_j, b_j}(x) z_{w_j, b_j}(x') = \hat{k}_D(x, x') \tag{47}$$

where $z_{w_j, b_j}(x) = \sqrt{2} \cos(w_j^T x + b_j)$ and $b_j \sim [0, 2\pi]$ uniformly. Random Fourier features approximations are unbiased and, in the specific case of a $d$-dimensional EQ kernel with $\lambda = 1$, we have to samples from the following Fourier transform:

$$g(w) = \left(\frac{2\pi}{\sigma^2}\right)^{-\frac{d}{2}} \exp\left(-\frac{\sigma^2 \|w\|_2^2}{2}\right) \tag{48}$$

which is a $d$-dimensional Gaussian distribution with zero mean and covariance matrix with all diagonal elements equal to $(1/\sigma^2)$.

The impact on the MMD from the use of random Fourier features to approximate the kernel for both the FWLS and FWLSBQ algorithms is demonstrated in Figure S3. In this example, the quadrature rule uses the kernel with random features but the MMD is calculated using the original $\mathcal{H}$-norm. The reason for using this $\mathcal{H}$-norm is to have a unique measure of distance between points which can be compared.

Clearly, we once again have that the rate of convergence of the FWLSBQ is much faster than FWLS when using the exact kernel. The same phenomena is observed for the method with high number of random features ($D = 5000$). This suggests that both the choice of design points and the calculation of the BQ weights is not strongly influenced by the approximation. It is also interesting to notice that the rates of convergence is very close for the exact and $D = 5000$ methods (atleast when $n$ is small), potentially suggesting that exponential convergence is possible for the exact method. This is not so surprising in itself since using a Gaussian kernel represents a prior belief that the integrand of interest is very smooth, and we can therefore expect fast convergence of the method.

However, in the case with a smaller number of random features is used ($D = 1000$), we actually observe a very poor performance of the method, which is mainly due to the fact that the weights are not well approximated anymore.

In summary, the experiments in this section suggest that the use of random features is a potential alternative for scaling up Bayesian Quadrature, but that one needs to be careful to use a high enough number of features. The experiments also give hope of having very similar convergence for infinite-dimensional and finite-dimensional spaces.

Figure S3: Random Fourier Features (RFF) for Bayesian Quadrature. RFF are used to approximate the EQ kernel in the example of the simulation study. The $\text{MMD}^2$ is plotted in the case where the EQ kernel is used (FWLS: blue; FWLSBQ: black), as well as when a using random features with $D = 1000$ (FWLS: red; FWLSBQ: purple) and $D = 5000$ (FWLS: orange; FWLSBQ: green).