[Reviews · NeurIPS 2015]

Submitted by Assigned_Reviewer_1

Update after response: with the clarifications mentioned, I'm very happy with the state of this paper as a theoretical contribution to an important problem, and am increasing my score from 7 to 8.

~~~

This paper unifies the Frank-Wolfe based approach for numerical integration with Bayesian Quadrature, developing a numerical integration scheme that provides a full posterior distribution over the result as well as giving good convergence rates. This then also tells us that BQ with optimally-selected points has at least the same convergence rates.

It seems that Theorem 1 says only that the BQ weights are no worse than FW weights, and then gives the same convergence rates as [2]. Thus this paper does not establish theoretically that FW(LS)BQ converges faster than FW(LS), but rather that it is no slower. This was not clear to me on first reading and perhaps should be noted more explicitly.

The assumption of a finite-dimensional Hilbert space may be standard, but it makes the results somewhat hard to interpret. This assumption seems to come in only through [2, Prop. 1]; [2, Prop. 2] establishes that this same convergence in fact does *not* hold for an infinite-dimensional RKHS. Though section 4.1 notes that any kernel is in practice finite-dimensional, it seems reasonable from the proof of [2, Prop. 1] that as the dimension of the kernel becomes large, the constants C will become quite small and convergence will become slow. It would thus be informative to include (perhaps in the appendix) experiments with a truly finite-dimensional kernel to see if the line search methods perform better in that case. One interesting such experiment might be to use approximations to the exponentiated-quadratic kernel (e.g. random Fourier features) of increasing dimension.

On line 279-280: This seems to imply that the main way kernel choice influences the convergence rate is through the constant R. But

R = \sup_{x, y \in X} ||\Phi(x) - \Phi(y)||

= \sup_{x, y \in X} \sqrt{k(x, x) + k(y, y) - 2 k(x, y)}. Consider the exponentiated-quadratic kernel of section 4.1. When X is unbounded, R = \sqrt{2} \lambda. One can thus minimize R by minimizing \lambda, but of course that is not the best path to take, and indeed changing \lambda also changes the constants C. In general, some discussion of how C is affected by the kernel choice would be quite interesting.

It seems that maintaining the constants ||f||, R, and so on in Theorems 1 and 2 would not be overly cumbersome; though C is of course difficult to determine, it at least satisfies a few simple scaling properties, and so this might provide added understanding in the behavior of the bound under different situations.

Lines 345-346 note that "the weights for SBQ are optimal at each iteration, which is not the case for FWBQ and FWLSBQ." Given the design points, don't FW(LS)BQ and SBQ weight those points in the same way?

It would be useful to briefly compare the computational costs of the various methods, both asymptotically (which is noted at various points through the paper but not fully compared) and in terms of actual runtimes.

Relatedly, it seems that M(n) must grow exponentially in n for the stated FWLSBQ rate to hold, which seems like a substantial computational burden.

The conjecture that the convergence rates also apply to SBQ seems reasonable. Given that, is there any practical situation in which FW(LS)BQ would be preferable to SBQ, or should it be used only by conservative practitioners seeking a guaranteed method?
Summary: The paper provides a new method for probabilistic numeric integration, and proves that it converges at least as quickly as previous methods while also providing quickly-contracting posteriors on the output. The method seems to be of mostly theoretical interest, however, since SBQ outperforms it empirically and is reasonably conjectured to have convergence rates at least as good as those of the new method.

Submitted by Assigned_Reviewer_2

The paper describes a method for numerical integration for the evaluation of expectations combining ideas from Bayesian Quadrature with convex optimisation techniques such as the FW algorithm. The main contribution is the derivation of theoretical guarantees that the method converges to the true value exponentially in the number of designed points and posterior contraction rates are super exponential.

The paper is very well written and clear for most parts. The technique is interesting and powerful. The main idea of connection FW with quadrature rules was presented in [2]. Here, the authors introduce these in the context of BQ and present theoretical guarantees in section 3. To this reviewer the paper contains most of the necessary elements for a solid NIPS paper. The main weaknesses in the paper are:

1) For such a general problem as computing expectations, the experimental section, with a simulated problem and one experiment on a practical problem is underwhelming. It would be interesting to see more examples and comparisons to other quadrature methods (e.g. Gauss-Hermite for the case where p(x) is Gaussian).

2) It wasn't clear how the method performs and scales with the input dimensionality. That's the main challenge with current quadrature methods; a few paragraphs on this would make the paper much stronger.

Summary: The paper presents a numerical integration approach combining Bayesian quadrature with convex optimisation techniques such as the FW algorithm. Convergence and contraction rates are provided. The method seems powerful and is nicely described. The main drawback is the experimental results that do not demonstrate the full potential of the method.

Submitted by Assigned_Reviewer_3

( Light review)

Nice work on an important and exciting area.
Summary: A technical paper bringing rigour to an area of interest to the NIPS community.

Submitted by Assigned_Reviewer_4

This paper a new Bayesian quadrature method based on Frank-Wolfe algorithm. The previous Bayesian quadrature has no theoretical results about convergence. The proposed method uses the Frank-Wolfe algorithm for picking the design points and computes the corresponding weights according to the Bayesian quadrature criterion. The performance of the proposed method is evaluated on sythetic data and a model selection problem on protein regulation data. The proposed method outperforms the previous Frank-Wolfe algorithms and reaches similar performance as sequential Bayesian quadrature.

Summary: This paper proposes a Frank-Wolfe algorithm based Bayesian quadrature method. The proposed method provides the theoretical retsults about the convergence rate.

Author Feedback
Author rebuttal: We gladly thank all six reviewers for their careful reading of our paper and welcome their positive and constructive suggestions. We have summarised and responded to the points of interest below:

--- REVIEWER 1 ---
(1) The suggestion to use random Fourier features to assess the performance of FWLSBQ under varying dimensionality of the RKHS is very interesting. We would be happy to run additional experiments and include them in revision.

On the general point of how rate constants depend on kernel choice, we completely agree that there is more that could be said here, and that lines 279-280 only provided part of this story. As shown in Bach et al. (2012), C is the radius of the largest ball centred on the mean element that is contained in the marginal polytope. In revision we will be sure to highlight the dependence of both C and R on kernel choice.

We are happy to make ||f|| and R explicit in the statement of Theorems 1 and 2 in revision.

(2) "Given the design points, don't FW(LS)BQ and SBQ weight those points in the same way?"

Apologies for confusion here. In FW(LS)BQ the standard FW(LS) weights are used at each iteration 1,...,n-1 and only at the very last step n are these changed for the BQ weights. The reason for this is that the Frank-Wolfe algorithm always requires non-negative weights at each intermediate step, which may not be the case if using BQ weights. We will make this clear in revision.

(3) On computational cost, the discussion of M(n) is mainly theoretical. In practice we would expect the next design point to be generated using a numerical optimisation procedure that is more sophisticated than optimisation by random sampling. However, for experiments in the paper we did employ random sampling because it fits elegantly into our theoretical analysis of FW(LS)BQ algorithms (see Lacoste-Julien et al., 2015, who also used random sampling in experiments). As such we did not focus on computational times for these experiments, which could be substantially improved by employing more intelligent approaches to optimisation, such as Bayesian Optimisation. We will act to clarify this point in revision.

(4) As you suggest, the main contribution of our work is to obtain a BQ method with theoretical guarantees; it is unclear to us whether there are situations in which FW(LS)BQ outperforms SBQ, which is known to be very powerful. Our hope is that the additional rigour provided by our work will lead to wider interest and further development of BQ methods in general, as these offer huge potential in scientific and engineering applications.

(5) Thank you for your suggestion on how to clarify some aspects of presentation. These will gladly be fixed.

--- REVIEWER 2 ---
(1) A comparison with other approaches to quadrature is currently underway. In fact, there are interesting links between BQ and existing quadrature rules, such as Gauss-Hermite, with the latter being derived from the former using a carefully constructed kernel function (see Saerkkae et al., 2015). Thus not just an empirical but a theoretical comparison with other approaches is possible. We originally decided not to discuss this in detail due to the space restrictions and the desire to not distract the reader from the main messages of the paper. However, we now intend to elaborate on this point in revision.

(2) High-dimensional integration remains profoundly challenging and our methods do not claim to resolve this problem. Most BQ methods employ transition invariant kernels and therefore inevitably suffer from a curse of dimensionality. There have been several attempt to improve scalability in the related field of Bayesian Optimization, such as the work of Wang et al. (IJCAI 2013) which works with random embeddings in lower-dimensional spaces. Similar techniques could potentially be used to improve scalability of BQ methods. We will be sure to discuss this in revision.

--- REVIEWERS 3,4,5,6 ---
Thank you for taking the time to read and comment on our submission. We will gladly act to clarify the aspects of presentation that you have highlighted in your feedback.